# Comparison of Activity in Scapular Stabilizing Muscles during Knee Push-Up Plus and Modified Vojta’s 3-Point Support Exercises

**DOI:** 10.3390/healthcare9121636

**Published:** 2021-11-26

**Authors:** Hyoungwon Lim

**Affiliations:** Department of Physical Therapy, Dankook University, Cheonan 31116, Korea; 11954690@dankook.ac.kr

**Keywords:** push-up plus, knee push-up plus, vojta therapy, scapular stabilizer muscles

## Abstract

Selective serratus anterior (SA) strengthening without compensatory movement of the shoulder stabilizers is essential for shoulder stability and functional movement without causing shoulder injury and dysfunction. The purpose of this study was to compare electromyographic (EMG) activity between the SA, upper trapezius (UT), lower trapezius (LT), and pectoralis major (PM) during the knee push-up plus (KPUP) and modified Vojta’s 3-point support (MV3PS) exercises. Scapular stabilizer muscle activity (UT, LT, SA, and PM) was investigated during the KPUP and MV3PS exercises in 40 healthy adults (19 males, 21 females) using surface EMG. Muscle activity of the SA was significantly higher during the MV3PS exercise than during the KPUP (*p* < 0.05). However, muscle activity in the PM was significantly lower during the MV3PS exercise (*p* < 0.05). In addition, the LT and UT showed less muscle activity during the MV3PS exercise, although the difference was not statistically significant (*p* > 0.05). These findings suggest that the MV3PS exercise better activates the SA than KPUP.

## 1. Introduction

Competitive sports, especially overhead sports, are physically demanding on the shoulder since they require a large range of motion and high-speed movements [1]. In some individuals, specific muscle imbalances have been implicated in the etiology of shoulder disorders such as scapular dyskinesis and shoulder impingement syndrome [2]. Scapular dyskinesia, which refers to abnormal mobility or function of the scapula, can result in muscle imbalance [3,4,5]. Impairments to the serratus anterior (SA) such as muscle imbalance or weakness often result in altered scapula positioning, scapulothoracic and humeral motion, and muscle activation patterns, which increase the risk of developing shoulder impingement, glenohumeral instability, pain, and other musculoskeletal disorders [6]. The SA is the primary muscle used to stabilize the medial border and inferior angle of the scapula and prevent winging and anterior tilting of the scapula [7,8]. Abnormal muscle firing patterns in the SA caused by muscle weakness or fatigue are associated with painful shoulder conditions such as subacromial impingement and scapular winging [9,10]. Most exercises recommended for activating the SA mainly include a protraction component, and the push-up exercise is known to be one of the most effective exercises for activating the SA [11]. In particular, the plus phase of the push-up exercise shows the highest degree of SA activation compared to other SA activation exercises [9,12]. Ludewig et al. [9] found that the SA was more selectively activated during push-up plus (PUP) exercises than standard push-up exercises. PUP is a closed kinetic chain (CKC) exercise, and unstable surfaces are often used during CKC exercises to increase the activation level of stabilizing muscles without drastically increasing muscle activity in agonists [2]. However, a previous study reported greater activation of the pectoralis compared to the SA in subjects with scapular wings during the plus phase of PUP [13]. Such excessive pectoralis major (PM) activation may lead to glenohumeral and scapulothoracic pathologies, such as shoulder anterior joint translations or decreased compressive forces on the glenoid fossa [14,15,16]. Therefore, selective SA activation without high PM activation is needed to reduce the likelihood of glenohumeral and scapulothoracic joint dyskinesias.

Vojta therapy is a treatment method developed by Dr. Vaclav Vojta, a pediatric neurologist from the Czech Republic, in which a systemic motor response in the body is prompted by specific peripheral stimuli in specific starting positions. The components of Vojta therapy include reflex turning and reflex creeping. The final motor response of the reflex creeping component is three-point support, which is supported by the medial epicondyle of the humerus, the ipsilateral medial femur condyle, and the heel on the contralateral side [17]. In Vojta therapy, each open kinetic chain and CKC exercise leads to a treatment response, which can be expressed as differentiation of muscle function [17]. This differentiation of muscle function occurs in CKC and also causes the coordinated contraction of various muscles through mechanical compression of the articular surface by strengthening muscles and increasing endurance [18]. There have been many recent studies on the clinical effects and treatment mechanisms of Vojta therapy [19,20,21,22,23]. This study hypothesized that three-point support, the final muscle response in Vojta therapy, would be effective for enhancing shoulder joint stability by prompting the coordinated contraction of various muscles and stimulating proprioceptors by triggering differentiation of muscle function through CKC.

Typically, clinicians utilize the push-up, PUP, and other push-up variations for retraining the SA, lower trapezius (LT), and upper trapezius (UT) muscles [24]. In addition, clinicians often recommend that female patients perform push-ups from a kneeling position to reduce the physical demands of the exercise by decreasing the long moment arms [25]. The knee-plus exercise is a more effective method for both selective activation of the SA and low activity of the UT than the knee push-up plus (KPUP) [25]. Maeo et al. [26] suggested that push-ups on an unstable surface require greater muscle activity in the proximal muscles to stabilize the upper limbs. However, other researchers have reported conflicting results, finding no difference in the muscle activity of the scapula stabilizing muscle when using an unstable surface [27,28,29,30]. In addition, studies have found that muscle activity in push-ups under different conditions was not related to exercise difficulty [11,24].

Therefore, the purpose of this study was to compare muscle activity in the scapular stabilizers when performing the KPUP and modified Vojta’s three-point support (MV3PS) exercises, both of which are recommended for scapulothoracic joint instability. The following hypotheses were formulated: (1) the MV3PS exercise will show higher SA muscle activity than the KPUP, and (2) the MV3PS exercise will show lower PM, UT, and LT muscle activity than the KPUP.

## 2. Materials and Methods

### 2.1. Subjects

This study was conducted with 40 healthy adults aged 20 to 29 years old who attended Dankook University. All study subjects voluntarily consented to the experiment after listening to an explanation of the study’s purpose and the exercise methods that would be used in the experiment. Once the participants provided written informed consent, the experiment was conducted. The criteria for the selection of study subjects were (1) those who did not have neck, shoulder, and upper extremity instability or musculoskeletal problems, (2) those who had not performed muscle strengthening exercises for the muscles around the shoulder within the previous 6 months, (3) those who did not have neurological problems, and (4) those who had not undergone shoulder surgery within the previous 6 months. Those who complained of pain during maximum voluntary isometric contraction (MVIC) were excluded from the experiment. This study was approved by Dankook University’s institutional review board (approval number DKU 2020-04-019).

### 2.2. Electromyographic Recording and Data Analysis

For surface electromyographic (EMG) measurements, the Desk DTS EMG system (Noraxon Inc., Scottsdale, AZ, USA, 2007) was used, and EMG signals were recorded to compare muscle activity while KPUP and MV3PS exercises were performed. The collected data were analyzed using Noraxon MyoResearch version 3.6. The sampling rate was set at 1000 Hz. A band pass filter was used between 20 Hz and 450 Hz. The raw data were processed to calculate the root mean square using a 50 ms window. MVIC was performed to normalize the EMG signals. In order to measure muscle activity, the electrode attachment locations for each muscle were determined based on previous studies [27,28]. The electrode attachment areas were prepared by abrading the area with fine sandpaper and cleansing it with alcohol. Some of the participants’ hair was shaved if necessary. Disposable Ag/AgCl surface electrode pairs were positioned at an interelectrode distance of 2 cm. Electrode attachment sites refer to where the surface electrodes were placed to measure muscle activity in the subjects’ UT, LT, SA, and PM muscles. The electrodes were attached to the upper extremity of the subjects’ dominant side, and the dominant side was the right side for every subject. In order to quantify the surface EMG values of the UT, LT, SA, and PM muscles, the MVIC of each muscle was measured while they were held in standardized manual muscle-testing positions. For the UT test, resistance was applied in a downward vertical direction with both shoulders raised from a sitting position. For testing the LT, the subject’s arm was placed at the height of their head along the direction of the muscle fibers while in the prone position, and the opposite side of the pelvis was gripped with one hand so that the pelvis did not lift. Then, the arm was placed on the distal part of the forearm with the opposite hand in the direction of the examination table while resistance was applied. To test the SA, the subject flexed his or her arm along the shoulder joint plane at a 120° angle in a sitting position and applied resistance in the direction of scapular retraction while protracting the scapula. To test the PM, resistance was exerted in the direction of horizontal abduction with the shoulder joint abducted at a 90° angle, the subject in the supine position, and the elbow joint flexed at a 90° angle, and the upper extremities were moved toward the chest to horizontal abduction with the shoulder joint [25,26]. The subjects maintained each position for 5 s, and in order to minimize persistent muscle fatigue that could occur during MVIC measurements, the subjects rested for 1 min after each measurement was taken. The MVIC of each muscle was measured three times, and after processing the root-mean-square data value for 5 s, the average EMG signal volume for 3 s, excluding the first and last seconds, was used as the muscle activity measurement (%MVIC).

### 2.3. Procedures

Before performing each exercise in this study, the subjects were familiarized with the KPUP and MV3PS postures for five minutes by listening to a thorough explanation, watching a video, and receiving a demonstration from the researcher. All subjects performed the two postures at random in order to exclude the order effect. In this study, the starting phase, holding phase, and ending phase were operationally defined in order to standardize exercise posture and performance under the two conditions and to control unnecessary movements. The participants each performed three trials of push-up exercises with a 1 min resting period between trials to minimize muscle fatigue. EMG signals were collected during the middle 3 s of the ending phase. The mean values of the three trials for each PUP exercise were used for data analysis. Data were not collected if the standardized starting phase, holding phase, and ending phase were not maintained.

#### 2.3.1. Knee Push-Up Plus

All trials were completed in a standardized, quadruped position with the hands shoulder-width apart, elbows fully extended, and shoulders flexed at a 90° angle during the exercise [20]. A target bar was placed at the level of the T-4 spinous process so that the subject could protract the scapula to the same height in order to maintain the same posture for each KPUP trial. In addition, the target line was positioned in the palm support position to prevent the body from moving forward excessively [27]. The KPUP phases were as follows: (a) the starting phase, during which the scapula was protracted by translating the thorax posteriorly until the spinous process of the thoracic vertebra touched the target bar for 2 s, (b) the holding phase, during which the position was held for 5 s, and (c) the ending phase, during which the subject returned to the starting position for 2 s. The duration of each exercise was controlled using an auditory signal generated by a metronome. When the subject’s T4 spinous process reached the target bar and the position of both shoulders was maintained without crossing the target line, the KPUP was considered to have been successfully performed (Figure 1).

#### 2.3.2. Modified Vojta’s 3-Point Support

In this study, the starting position of the MV3PS exercise was modified to be a four-point support posture so that the subject’s weight was maintained on both the medial epicondyle of the humerus and the medial femur condyle in order to be in the quadruped position, which is the same position subjects assumed when performing the KPUP exercise. The target bar and target line were also placed in the same manner as the KPUP. Other procedures were also performed in the same manner as the KPUP. The phases of the MV3PS exercise were as follows: (a) the starting phase, during which the scapula was protracted by translating the thorax posteriorly until the spinous process of the thoracic vertebra touched a target bar for 2 s, (b) the holding phase, during which the subject maximized extension of the ipsilateral wrist and held the position for 5 s, and (c) the ending phase, during which the subject returned to the starting position for 2 s (Figure 1).

### 2.4. Statistical Analysis

Statistical analysis was performed using SPSS for Windows version 21.0 (IBM Inc., Armonk, NY, USA). The paired *t*-test was performed to compare differences in the muscle activity of the UT, LT, SA, and PM muscles when the KPUP and MV3PS exercises were performed. The level of significance was set at α = 0.05. Cohen’s *d* was calculated to determine the effect size of the results.

## 3. Results

### 3.1. General Characteristics of the Subjects

The descriptive characteristics of the subjects can be found in Table 1.

### 3.2. Comparison of Muscle Activity between Scapular Stabilizers during KPUP and MV3PS

The MV3PS exercise showed significantly higher muscle activity in the SA than the KPUP (*p* < 0.05). However, muscle activity in the PM was significantly lower during the MV3PS exercise than the KPUP (*p* < 0.05). In addition, muscle activity in the LT and UT during the MV3PS exercise was lower than during the KPUP, but the difference was not statistically significant (*p* > 0.05). The effect size of each muscle was 0.720 for the PM, 0.175 for the LT, 0.678 for the SA, and 0.024 for the UT (Table 2) (Figure 2).

## 4. Discussion

This study was based on Vojta therapy, and the main findings were that muscle activity in the SA was higher during the MV3PS exercise than during the KPUP, while muscle activity in the PM was lower during the MV3PS exercise. In addition, muscle activity in the UT and LT was lower during the MV3PS exercise than the KPUP, but the difference was not statistically significant. In this study, the effect size was calculated using Cohen’s *d*, and the medium effect size was 0.720 and 0.678 in the PM and SA muscles, respectively.

In a previous study, exercises involving low PM activity to promote selective SA activation and reduced risk of glenohumeral joint pathology were found to be an important factor in rehabilitation [28]. In addition, high muscle activity in the PM increased the anterior translation of the humerus and caused instability of the anterior joint capsule [8]. Another study reported that, when the PUP was performed in a standardized position, less scapular protraction due to decreased SA activity may have caused an increase in PM activity to achieve the same range of motion using additional clavicle protraction and humeral translation [28]. Another previous study reported that loads greater than 35% of a person’s body weight were exerted on a single upper extremity during the standard PUP [30]. De Mey et al. [2] reported that muscle activity in the SA decreased and muscle activity in the PM increased when a knee prone bridging exercise similar to the exercise posture used in this study was performed on an unstable surface sling (a Redcord sling; RS). The previous study argued that the PM was the only muscle with a significant increase in muscle activity levels due to the RS. The study explained, first, that low muscle activity in the SA was a response to the unstable surface and, second, that the hand was in a relatively high position, which placed less weight on the upper extremities than the lower extremities [2]. This finding is consistent with the results of the present study. Although this study did not use an unstable surface, the MV3PS exercise features a lower hand position than the KPUP, so more weight can be applied on the upper extremities during the MV3PS exercise than the KPUP. Therefore, the low position featured in the MV3PS exercise could explain the increase in muscle activity in the SA. Maeo et al. [26] reported that trunk muscle activity during instability exercises increased as the difficulty of the task increased. However, other studies have reported results that were not affected by the difficulty of push-up exercises under various conditions [11,24]. Castelein et al. [11], for example, examined muscle activity in the SA and pectoralis minor under different exercise conditions. The highest muscle activity in the SA was found when subjects performed the serratus punch without upper extremity weight-bearing. The pectoralis minor also showed high muscle activity during the serratus punch, but there was no difference in muscle activity compared to the modified PUP floor version in which weight was applied to the upper extremity [11]. Horsak et al. [24] suggested that the knee-plus exercise, which is relatively less difficult to perform, is an effective method for inducing isolated activation of the SA and low activity in the UT compared to the KPUP. In this study, despite differences in exercise difficulty, unlike previous studies, an ideal result was obtained in that SA activity increased and PM activity decreased. For this reason, during the plus phase of the KPUP, the weight load of the upper extremity is distributed to the hands, elbows, and shoulders, whereas during the plus phase of the MV3PS exercise, the weight load is believed to be concentrated on the elbow and shoulder joints, increasing the need for shoulder muscles to be used. Contreras et al. found that the percentage of body mass supported by changes in moment arm flexion during push-ups was 75% in the down position and 69% in the up position [31]. Therefore, it is believed that the MV3PS exercise requires greater force in a position relatively closer to the floor than the KPUP, and the increased difficulty of the task may have led to higher core muscle activity.

Gajewska et al. [32] conducted a study to explain the mechanism of Vojta therapy in healthy subjects, in which muscle activation was observed in the contralateral deltoid, ipsilateral deltoid, and contralateral rectus femoris muscles after stimulation of the lower femoral epicondyle. Their study reported that stimulation of the acromion and femoral epicondyle led to cross-excitation of spinal centers that mainly coordinated the muscles of the lower and upper extremities. Ha and Sung [23] reported improvements in trunk stability and gait after Vojta therapy was conducted for children with cerebral palsy. The MV3PS exercise was thought to induce reflex creeping in Vojta therapy, and it was modified to suit the needs of this study. The plus phase of this CKC involved posterior translation of the thorax in a relatively fixed scapula. In Vojta therapy, the plus phase is enabled through the support of the elbow on the face and the heel on the occipital side. The MV3PS exercise was a CKC with the thoracic wall moving relative to the arm. This was a modification of the expected response based on Vojta therapy. In the plus phase, posterior translation is performed for a relatively fixed scapula. This is possible through both the medial epicondyle of the humerus and the medial condyle of the femur. At this time, the scapula on the side of the elbow support covers the humerus head, which is possible through activation of the SA, and muscle contraction in the PM is reversed through elbow support. In other words, especially in the MV3PS exercise, PM muscle action is reversed as the elbow is supported despite the difference in exercise mode and body position (hand vs. elbow support). This is called differentiation of muscle function in Vojta therapy [17]. The bearing medial epicondyle of the humerus causes eccentric contractions in the PM, which lift the trunk off the ground. Therefore, the MV3PS exercise features a narrower base of support than the KPUP, which leads to a more dynamic posture and requires trunk stabilization. Push-ups and their variations require a certain level of strength in the upper extremity in addition to well-developed trunk stability [24]. Therefore, it is thought that the increase in upper extremity weight-bearing through eccentric contraction in the PM and elbow support increases the demands on the scapula muscle, resulting in increased muscle activity in the SA and low muscle activity in the PM. Ebben et al. [33] examined the maximum ground reaction forces (GRFs) according to various push-up conditions (regular, flexed knee, with the feet elevated on 30.48 cm and 60.96 cm boxes, and with the hands elevated on 30.48 cm and 60.96 cm boxes) to quantify exercise load. High GRFs were found for push-ups with the feet elevated, whereas low GRFs were found for push-ups with the hands elevated in the knee flexion state [33]. A greater maximum GRF indicates that more muscles were recruited. Therefore, in the present study, the MV3PS exercise, since it had a lower upper limb height than the KPUP, produced a higher GRF, which is believed to have resulted in more muscle recruitment. In this study, muscle activity in the UT and LT during the MV3PS exercise was not statistically significant, though it was found to be low. Horsak et al. [24] found that the KPUP slightly increased muscle activity in the UT and LT compared to knee-plus exercises due to increased physical demands for flexion and extension of the elbow joint and stabilization of the scapulothoracic and shoulder joint complex. The fact that peak EMG amplitudes were lower for comparable exercises suggests that differences are primarily due to the lower level of force used in this study [34]. Average amplitude is a quantitative measure of electrical activity relative to a specific phase of movement [34]. Although the relationship between EMG activity and force is multifactorial, during isometric conditions, “almost without exception, investigators report either linear relationships or a more than linear increase of the EMG signal with increasing force” [35]. Therefore, the KPUP increased muscle activity in the UT and LT, stabilizing the scapulothoracic and shoulder joint complex, which the MV3PS exercise did not. Martins et al. [36] found that excessive UT activity was a characteristic of individuals with weakness of the SA, which corresponds to a high risk of developing impingement syndrome and shoulder pain. The SA is a primary agonist for widening the subacromial space to prevent impingement of the underlying tissue [37]. In a study comparing plus exercise under different conditions, decreases in UT muscle activity and increases in LT muscle activity were not statistically significant without RS conditions during a knee prone bridging plus exercise, which was similar to the exercises featured in this study [24]. In addition, SA muscle activity increased and PM muscle activity decreased in non-RS conditions, which is consistent with the results of this study [24]. This was likely due to the larger glenohumeral muscles, such as the PM, compensating for the smaller scapulothoracic muscles such as the SA, LT, and UT, which acted as stabilizing muscles [24]. Therefore, the MV3PS exercise requires a higher level of force than the KPUP, and it may be possible to activate the muscles around the scapulothoracic joint rather than the muscles around the glenohumeral joint according to differences in the position of the body.

Selective SA activation without high PM activation is an important factor for reducing the likelihood of glenohumeral and scapulothoracic joint dyskinesia [28]. Therefore, the MV3PS exercise, with low PM muscle activity and non-statistically significant but low muscle activity in the UT and LT, would be an important factor in rehabilitation to promote selective SA muscle activation and reduce the risk of glenohumeral joint pathology.

A limiting factor to be aware of when interpreting the present results is that the results of this study cannot be generalized to a population of other age groups with different symptoms since our study sample was limited to a young population without scapular winging. Second, the pectoralis minor is also known to be closely related to the topic of this study since it is attached to the anterior chest wall. Therefore, other muscles should be examined in future studies. Third, it was not possible in this study to quantify the difficulty of different exercises. In future studies, the exercise load of subjects with shoulder pathology should be quantified, and the methodology of this study should be expanded to various muscles, exercise modes, and exercise conditions.

## 5. Conclusions

This study was based on Vojta therapy, and the following results were obtained. The MV3PS exercise showed greater muscle activity in the SA and lower muscle activity in the PM than the KPUP. In addition, although muscle activity in the LT and UT was not statistically significant, it was lower than that of the KPUP during the MV3PS exercise, which was consistent with the hypothesis of the study. Therefore, the MV3PS exercise is recommended for healthy subjects as an optimal exercise when maximum SA activation and minimal PM activation is required.

## Figures and Tables

**Figure 1 healthcare-09-01636-f001:**
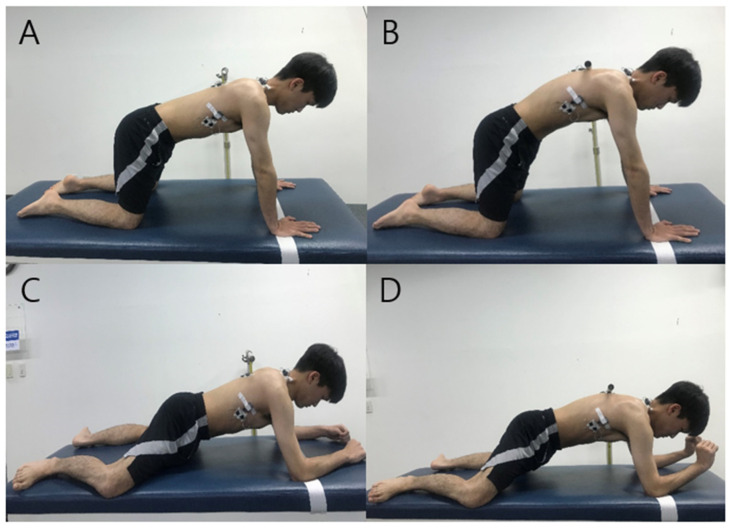
Knee push-up plus ((**A**) starting position; (**B**) ending position) and modified Vojta’s 3-point support ((**C**) starting position; (**D**) ending position) positions.

**Figure 2 healthcare-09-01636-f002:**
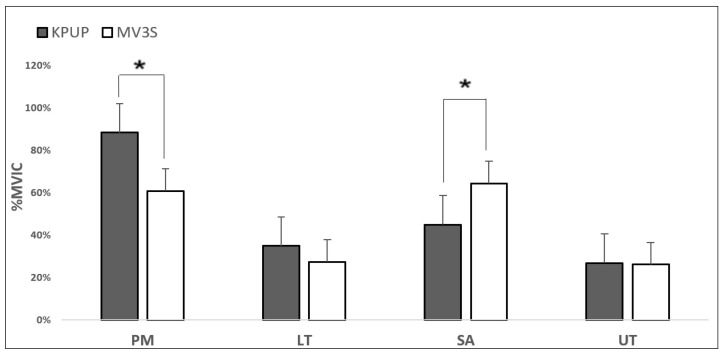
Comparison of shoulder stabilizer muscle activity during exercises. PM, pectoralis major; LT, lower trapezius; SA, serratus anterior; UT, upper trapezius; KPUP, knee push-up plus; MV3PS, modified Vojta’s 3-point support. * *p* < 0.05.

**Table 1 healthcare-09-01636-t001:** General characteristics of subjects (N = 40).

Gender	Number of Subjects	Age (Years)	Height (cm)	Weight (kg)
Male	19	20.86 ± 2.31	168.00 ± 7.13	63.9 ± 11.50
Female	21	21.02 ± 2.24	167.80 ± 7.21	62.6 ± 10.50

**Table 2 healthcare-09-01636-t002:** Comparison of scapular stabilizer muscle activity (%MVIC) during exercises.

Muscle	KPUP	MV3PS	*p*	Effect Size
	Mean ± SD	Mean ± SD		
PM	88.28 ± 11.40	60.76 ± 9.76	<0.001 *	0.720
LT	34.91 ± 6.95	27.43 ± 3.25	0.276	0.175
SA	44.88 ± 3.69	64.36 ± 6.04	<0.001 *	0.678
UT	26.86 ± 5.44	26.10 ± 4.29	0.880	0.024

MVIC, maximum voluntary isometric contraction; PM, pectoralis major; LT, lower trapezius; SA, serratus anterior; UT, upper trapezius; KPUP, knee push-up plus; MV3PS, modified Vojta’s 3-point support. * *p* < 0.05.

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
