# Peer review of "Comparison of Activity in Scapular Stabilizing Muscles during Knee Push-Up Plus and Modified Vojta’s 3-Point Support Exercises"

_healthcare, 2021, doi:10.3390/healthcare9121636_

Round 1
Reviewer 1 Report
Line 18: remove "more efficient scapular stabilizing exercise" since it only improves activation of the SA
Line 53: can you provide more detail on Vojta therapy, what is it mainly used for and how does it relate to shoulder disfunction?
In Section 2.3: spell out the numbers (2 should be two)
Line 173: dorsiflexion should be extension
Photos in Figures 1 and 2 are switched.
Line 298: You mention GRF in the present study, but it looks like you did not collect GRFs. Please clarify.
Author Response
Line 18: remove "more efficient scapular stabilizing exercise" since it only improves activation of the SA
>>> corrected
Line 53: can you provide more detail on Vojta therapy, what is it mainly used for and how does it relate to shoulder disfunction?
>>> Vojta therapy is being used extensively in clinical settings and there are various ongoing clinical trials on Vojta therapy. As mentioned in the paper, the idea of this research has been initiated from the use of Vojta therapy. As stated above, when a patient receives Vojta therapy, functional differentiation occurs in muscles. This is possible in the CKC environment, and 3-point supports can be explained by the differentiation of the muscle function of pectoralis major. In order to explain the details of Vojta therapy, we have to go into extreme and tedious clinical details which I believe are almost impossible to explain briefly.
In Section 2.3: spell out the numbers (2 should be two)
>>> corrected
Line 173: dorsiflexion should be extension
>>> corrected
Photos in Figures 1 and 2 are switched.
>>> corrected
Line 298: You mention GRF in the present study, but it looks like you did not collect GRFs. Please clarify.
>>> I did not collect GRF in this study but referred to other studies that measured GRF in order to explain the results.
Reviewer 2 Report
The article is overall well written and well organized.
I would suggest to improve the quality of the manuscript as follows:
1) INTRODUCTION, lines 62-66: "This study hypothesized that 3-point support, the final muscle response in Vojta therapy, etc". I would try to rephrase. Are you referring to your study or to the "recent studies"? You should specify it, also because your study is about a 4-point support modified version;
2) MATERIALS AND METHODS: the "Electromyographic Recording and Data Analysis" is a bit text heavy, but I know how difficult could be to explain a procedure like that in an easy way.
Lines 130-135 and lines 143-147: since these lines nearly talk about the same things, and since, as I told before, the "Electromyographic Recording and Data Analysis" is a bit text heavy, could be possible to talk about the performance and the data analysis only in the "Procedures" sub-paragraph?
3) MATERIALS AND METHODS, "Procedures" 2.3.1 and 2.3.2. I strongly suggest to put the figures related to each exercise under the section in which it is explained how are them performed.
Furthermore, there is a mistake with the number and legends of Figures at page 6: Figure 1 has the legend of Figure 2 and viceversa. Fix them and be sure that along the manuscript each figure is correctly mentioned.
4) RESULTS: well and easy described, just an issue in Table 2: KPUP, Mean±SD, PM, and 88.28±11.40 are in bold. Did you intentionally put it in bold or it is a graphic mistake?
5) DISCUSSION: a bit text heavy as well, but well written. Well done with the study's limitations and suggestions for further researches. At line 331, I would suggest to use "would be an important factor in rehabilitation..." instead of "will".
Author Response
1) INTRODUCTION, lines 62-66: "This study hypothesized that 3-point support, the final muscle response in Vojta therapy, etc". I would try to rephrase. Are you referring to your study or to the "recent studies"? You should specify it, also because your study is about a 4-point support modified version;
>>> this study, hypothesized that Modified Vojta's 3-Point Support would be effective for enhancing shoulder joint stability by prompting the coordinated contraction of various muscles and stimulating proprioceptors by triggering differentiation of muscle function through CKC.
2) MATERIALS AND METHODS: the "Electromyographic Recording and Data Analysis" is a bit text heavy, but I know how difficult could be to explain a procedure like that in an easy way.
Lines 130-135 and lines 143-147: since these lines nearly talk about the same things, and since, as I told before, the "Electromyographic Recording and Data Analysis" is a bit text heavy, could be possible to talk about the performance and the data analysis only in the "Procedures" sub-paragraph?
>>> Yes, I agree with the reviewers. However, I have described all of them because I think there are some subtle differences in the procedure for each exercise.
3) MATERIALS AND METHODS, "Procedures" 2.3.1 and 2.3.2. I strongly suggest to put the figures related to each exercise under the section in which it is explained how are them performed.
>>> CORRECTED
Furthermore, there is a mistake with the number and legends of Figures at page 6: Figure 1 has the legend of Figure 2 and viceversa. Fix them and be sure that along the manuscript each figure is correctly mentioned.
>>> CORRECTED
There were no issues in the original file. However, it seems that there was a formatting error while uploading the file which caused the switch in lines.
4) RESULTS: well and easy described, just an issue in Table 2: KPUP, Mean±SD, PM, and 88.28±11.40 are in bold. Did you intentionally put it in bold or it is a graphic mistake?
>>> CORRECTED
5) DISCUSSION: a bit text heavy as well, but well written. Well done with the study's limitations and suggestions for further researches. At line 331, I would suggest to use "would be an important factor in rehabilitation..." instead of "will".
>>> CORRECTED